# *Saccharomyces boulardii*: What Makes It Tick as Successful Probiotic?

**DOI:** 10.3390/jof6020078

**Published:** 2020-06-04

**Authors:** Pedro Pais, Vanda Almeida, Melike Yılmaz, Miguel C. Teixeira

**Affiliations:** 1Department of Bioengineering, Instituto Superior Técnico, Universidade de Lisboa, 1049-001 Lisboa, Portugal; pedrohpais@tecnico.ulisboa.pt (P.P.); vanda_pintalmeida@hotmail.com (V.A.); melikeyilmaz@tecnico.ulisboa.pt (M.Y.); 2Biological Sciences Research Group, IBB-Institute for Bioengineering and Biosciences, Instituto Superior Técnico, 1049-001 Lisboa, Portugal

**Keywords:** *Saccharomyces boulardii*, *Saccharomyces cerevisiae*, probiotics, gastrointestinal tract

## Abstract

*Saccharomyces boulardii* is a probiotic yeast often used for the treatment of GI tract disorders such as diarrhea symptoms. It is genetically close to the model yeast *Saccharomyces cerevisiae* and its classification as a distinct species or a *S. cerevisiae* variant has long been discussed. Here, we review the main genetic divergencies between *S. boulardii* and *S. cerevisiae* as a strategy to uncover the ability to adapt to the host physiological conditions by the probiotic. *S. boulardii* does possess discernible phenotypic traits and physiological properties that underlie its success as probiotic, such as optimal growth temperature, resistance to the gastric environment and viability at low pH. Its probiotic activity has been elucidated as a conjunction of multiple pathways, ranging from improvement of gut barrier function, pathogen competitive exclusion, production of antimicrobial peptides, immune modulation, and trophic effects. This review summarizes the participation of *S. boulardii* in these mechanisms and the multifactorial nature by which this yeast modulates the host microbiome and intestinal function.

## 1. Introduction

Probiotics are defined as live organisms which, when administered in adequate amounts, confer a health benefit to the host, independently of where the action takes place and of the type of administration. They are normally recommended to help strengthen host systems, for example the gastrointestinal (GI) tract, and assist in the recovery of certain diseases. According to this definition, probiotics in food must contain at least 10^6^ CFU/g of viable and active microorganisms, while freeze-dried supplements have shown good results with 10^7^ to 10^11^ viable microorganisms per day [1,2,3,4,5]. It is also preferable that these are of human origin and that they cannot transfer any antibiotic resistance, pathogenicity or toxicity factors [4].

The most commonly used probiotics comprise lactic acid producing bacteria (*Lactobacillus* spp., *Bacillus* spp., *Bifidobacterium* spp., *Streptococcus* spp., and *Enterococcus* spp.) that are found in the human gastrointestinal tract, usually ingested in fermented foods [4]. These probiotics can be used by themselves or combined with each other, although it should be noted that not all combinations are stable and different strains of the same probiotic bacteria can have different capabilities or enzymatic activities, even if they belong to the same species [4,6]. Probiotic properties widely differ between species, strains or even between strain variants, which means these properties can be strain/variant-specific [4].

The ability of a given organism to display probiotic activity is also dependent on its ability to compete for a host niche. Probiotics must compete with pathogens that adhere specifically to host cells, such as those of the GI tract, including *Helicobacter pylori* or *Clostridium difficile*, but also *Borellia* spp., *Treponema* spp. or *Spirilium* spp. [2]. This means that the competition between probiotic microorganisms and pathogens is dependent on habitat-related idiosyncrasies [2]. Host factors can also influence the effectiveness of a probiotic. Genetic factors, baseline immune functions or microbiome diversity vary among individuals, which together with environmental factors (e.g., diet or stress) account for unique backgrounds where the same probiotic will have distinct outcomes [4].

Several bacteria have been identified as probiotics and their modes of action scrutinized to some extent, but yeasts may also exhibit probiotic properties. The baker’s yeast *Saccharomyces cerevisiae* does not seem to present significant advantageous attributes for human health [1]. On the other hand, the closely related *Saccharomyces boulardii* is effective in complementing the treatment of acute gastrointestinal diseases such as diarrhea or chronic diseases such as inflammatory bowel disease (IBD) [7,8]. To date, this is the only yeast used as a probiotic [4] and its probiotic properties are supported by scientific evidence from the *S. boulardii* CNCM I-745 (or *S. boulardii* Hansen CBS 5926) strain produced by Laboratoires Biocodex, highlighted by more than 80 randomized clinical trials [1]. Nevertheless, the efficacy of this strain cannot be extrapolated to other strains, like *S. boulardii* CNCM 1079 [1]. 

In this review, current knowledge on *S. boulardii* traits that support its probiotic nature and the correlation with distinctive features when compared with the non-probiotic *S. cerevisiae* will be explored. Focus will be given on reviewing the biology, genetics, ability to colonize the human gut and compete with gastrointestinal pathogens as features that may underlie the probiotic activity of *S. boulardii*. Unanswered questions, mostly related to the genetic basis underlying the probiotic phenotype, are discussed.

## 2. *S. boulardii* and *S. cerevisiae*: Similar but Different

The budding yeast known as *S. boulardii* is usually referred to as a distinct species within the *Saccharomyces* genus, despite being genetically close and sharing a similar karyotype to the model yeast *S. cerevisiae* [9,10,11]. Molecular typing studies resorting to pulsed-field gel electrophoresis (PFGE), randomly amplified polymorphic DNA-polymerase chain reaction (RAPD-PCR), and restriction fragment length polymorphisms (RFLP) of non-transcribed spacer (NTS) or internal transcribed spacer (ITS) reveal that *S. boulardii* strains from distinct origins all belong to a clearly delimited cluster within the *S. cerevisiae* species, arguing that they should be considered different strains of the same species [10,12]. Likewise, a DNA/RNA hybridization spotted microarrays study also concluded that *S. boulardii* is a strain of *S. cerevisiae* that has lost all intact Ty1/2 elements rather than a different species [13], while another study identified Ty1/3/4 as absent elements, but not Ty2/5 [11]. Phylogenetic analysis also shows that *S. boulardii* clusters are closely related to *S. cerevisiae* wine strains [11]. In spite of such similarities, microsatellite polymorphisms may provide a way to differentiate both species and identify *S. boulardii* properly [14,15].

Despite the striking relatedness in molecular phylogeny and typing, *S. boulardii* does possess identifiable distinct traits and is physiologically and metabolically distinct from *S. cerevisiae* (Table 1). Namely, *S. boulardii* is incapable of producing ascospores, switching to haploid form, or using galactose as carbon source [11,16,17,18,19]. It is more resistant to temperature and acidic stresses, but less resistant to bile salts [12,18].

## 3. *S. boulardii* Genomic Variations Provide Hints for Its Physiological Properties

*S. boulardii* and *S. cerevisiae* genomes were found to differ in internal regions of lower copy number in three chromosomes: chromosome I (*PRM9*, *MST28*, *YAR047C*, *YAR050W*, *CUP1*, *YAR060W* and *YAR061W*); chromosome VII (*YGL052W* and *MST27*) and chromosome XII (*ASP3* and *YLR156W*). *PRM9*, *MST27* and *MST28* genes encode nonessential membrane proteins specific to the *Saccharomyces sensu stricto* species [18]. *YAR050W* encodes a lectin-like protein that participates in flocculation; Asp3 is a nitrogen catabolite-regulated cell wall L-asparaginase II. *CUP1* had a two times lower number of copies than the average for *S. cerevisiae* species, possibly causing the increased sensitivity to copper in *S. boulardii* when compared to other *S. cerevisiae* strains [18].

Within genes with higher copy number, two functions are well represented: protein synthesis (*RPL31A*, *RPL41A*, *RPS24B*, *RPL2B* and *RSA3*) and stress response (*HSP26*, *SSA3*, *SED1*, *HSP42*, *HSP78* and *PBS2*). It is possible that these genes aid in increased growth rate and pseudo-hyphal switching and in higher resistance to high pH [18]. Duplicated and triplicated genes mostly encode stress response proteins, elongation factors, ribosomal proteins, kinases, transporters and fluoride export, which might aid in *S. boulardii* adaptation to stress conditions [11]. Altered gene copy number and mutations when compared to *S. cerevisiae* in the *SDH1* and *WHI2* genes was associated with increased acetic acid production by *S. boulardii*, correlated with antimicrobial activity [22].

*S. boulardii* was shown to display enhanced ability for pseudo-hyphal switching during nitrogen starvation compared to other *S. cerevisiae* strains [18]. Several genes related to pseudo-hyphal growth have considerably different number of copies: *CDC42*, *DFG16*, *RGS2*, *CYR1* and *CDC25* have higher copy number; *STE11*, *SKM1* and *RAS1* have lower copy numbers [18]. Some of these genes are involved in cyclic adenosine monophosphate (cAMP) pathways, suggesting its hyperactivation can lead to increased pseudo-hyphal growth. As a possible consequence, *S. boulardii* ability to create pseudo-hyphae was observed to be faster and more extensive than *S. cerevisiae* [18]. 

Variations in the number of repetitive sequences within flocculation genes was also identified in *S. boulardii*, namely in *FLO1*. The encoded flocculin was found to harbor additional copies of residue repeats when compared with most *S. cerevisiae* strains [11]. The Flo8 protein was also found to differ between *S. boulardii* and some *S. cerevisiae* strains where a point mutation results in a truncated protein (including in the reference strain S288c), resulting in defective flocculation and adhesion [23]. Other flocculation genes (*FLO10* and *FLO11*) detected in *S. boulardii* were not found to harbor differences in the copy number and period length of the repeats [11]. The higher maximum number of repeats (e.g., *FLO1*) in *S. boulardii* may affect its adhesion and flocculation ability, as well as sensitivity to stress [11]. 

Several studies have shown that *S. boulardii* is unable to use galactose as a carbon source, despite harboring all galactose uptake and fermentation genes [16,17,19]. Some studies have proposed that it is able to assimilate, but not ferment, galactose, possibly due to energy requirements [24,25]. More recently, a mutation in the gene *PGM2* was also associated with the inefficient use of galactose [17]. *S. boulardii* is also unable to use palatinose, possibly related with the absence of 3 isomaltase encoding genes (*IMA2*, *IMA3* and *IMA4*), which is involved in palatinose uptake and metabolism [11,19].

## 4. Adaptation to Host Environment

Probiotics must be able to endure in adverse conditions. *S. boulardii* optimal growth temperature corresponds to the human host temperature (37 °C), while *S. cerevisiae* grows optimally at 30 °C. *S. boulardii* is also more resistant to very high temperatures keeping 65% viability after one hour at 52 °C, while *S. cerevisiae* loses viability down to 45% [12].

The main obstacles in the stomach are the very acidic pH (2 to 3) and the presence of proteases such as pepsin that kill most microorganisms, including probiotics that enter the organism [26]. Diseases like hypochlorhydria decrease the bactericide properties of the stomach and make the patient more susceptible to infections by *H. pylori* and *Salmonella* spp. and to migrations of pathogenic microorganisms to the small intestine where they establish themselves [26]. In the case of the small intestine, main stressors include the high concentrations of bile salts, pancreatic enzymes, hydrolytic enzymes, pancreatin, organic acids, the integrity of the epithelial and brush border, the immune defense and the native microbiome and its secondary metabolism products (H_2_S, bacteriocins, organic acids) [27]. Bile salts are detergents produced in the liver from cholesterol and secreted to the intestine to improve nutrient absorption. As detergent like molecules, bile salts can be toxic to GI tract microorganisms by disrupting their cellular membrane lipid bilayer structures [12]. However, some probiotics are able to resist degradation by hydrolytic enzymes and bile salts [6]. For example, *S. boulardii* and *Bacillus coagulans* remain viable after exposure to simulated gastric juice containing pepsin and hydrochloric acid. These probiotics were also seen to be stable to the impact of bile salts [6]. *Bacillus clausii* was partially resistant to these conditions [6]. On the other hand, most *Lactobacillus* and *Bifidobacterium* spp. have reduced viability under exposure to gastrointestinal agents such as pepsin, hydrochloric acid and bile [6].

In vitro testing of probiotic formulations consisting of *S. boulardii* and bacterial probiotics (*Lactobacillus* spp. and *Bifidobacterium* spp.) highlighted the ability of *S. boulardii* to survive GI tract conditions. *S. boulardii* was able to survive after incubation in a gastric-like environment and in an intestinal environment (bile salts, pancreatin, pH 7.0) for 3 h, whereas the viability of the bacterial probiotics was severely impaired [6]. *S. boulardii* is also more resistant to a gastric environment than *S. cerevisiae*, while the viability of both species in an intestinal environment (sodium chloride, pepsin, pancreatin, pH 8.0) is not affected after 1 h [12]. Accordingly, 1 h was enough to show that *S. boulardii* is more resistant to low pH than *S. cerevisiae*, particularly at pH 2.0 [18]. Interestingly, although *S. boulardii* can survive the GI environments, its viability is significantly increased for 2 h if encapsulated by a double layer with sodium alginate and gelatin [28]. Tolerance displayed by *S. boulardii* to bile salts has also been tested. Surprisingly, *S. cerevisiae* is more tolerant to bile salts than *S. boulardii*. However, after 1 h, both species show a tolerance threshold bellow what would be considered as resistance to bile salts [12]. 

Dynamic modelling of the stomach and small intestine conditions also showed *S. boulardii* to be resilient to gastric and lower intestinal conditions, while modelling of the colon environment revealed the yeast is not able to colonize the colon, but had an individual-dependent effect in the microbiotic profile [29]. Other studies also point to the inability of *S. boulardii* to colonize the gut, suggesting that this yeast does not strongly adhere to intestinal epithelial cells and is quickly removed from the gastrointestinal system in healthy individuals [18]. However, it has been shown to colonize the intestine of gnotobiotic mice after a single administration [21]. This may mean that, although *S. boulardii* can colonize the gut, competition with intestinal microbiome is limiting [18]. Indeed, both *S. boulardii* and other *Saccharomyces* strains were shown to be unable to remain attached to human and mouse epithelial cells, in vitro and in vivo, respectively [18]. However, they do adhere to Caco-2 cells through an extracellular factor, probably secreted mucus [18]. Colonization of the gut was observed to be dependent, both in mice and human, on repeated administration over several days [20,21,30]. Moreover, administration of ampicillin increased *S. boulardii* cell concentration [20], reinforcing the notion that competition with intestinal microbiome plays a relevant role in the establishment of this yeast.

## 5. Mechanisms of Action

The gut microbiome is responsible for a multitude of roles, including protection against pathogen colonization, epithelial barrier maintenance or modulation of immune activity [31]. The mechanisms by which gut microbiome homeostasis is maintained are not yet fully understood. Probiotics are believed to display a variety of mechanisms: antitoxin effects, physiological protection, modulation of the normal microbiome, metabolic regulation and signaling pathway modification, nutritional and trophic effects, immune system regulation, pathogen competition, interactions with the brain-gut axis, cellular adhesion, cellular antagonism and mucin production [1,4,31]. *S. boulardii* has been described as participating in a number of these effects as part of its probiotic activity (Figure 1). The genetic basis and mechanistic details that underlie these observations are not fully understood and their clarification could be key to better exploit this yeast and how to potentiate general probiotic activity.

### 5.1. Modulation of The Normal Microbiome

Modulation of the normal microbiome may be favored directly by transiting probiotics which produce antimicrobial substances, or indirectly contribute to immune modulation or gut barrier function [31]. The use of probiotics has typically been applied to reestablish the normal gut microbiome upon dysbiosis. Gut dysbiosis refers to changes in the microbiome’s quantitative and qualitative composition. These changes may contribute to a disease state frequently associated to inflammation and can be a result of antibiotic-associated diarrhea, acute infectious diarrhea or IBD [31,32]. Probiotics treatment helps to stabilize the gut microbial community and lead to an improved disease outcome [32]. While some probiotics may become a part of the microbiome, others simply pass through the GI tract and modulate or influence the existing microbiome before exiting the body [31].

Several factors can have deleterious effects on the gut microbiome and hinder its protective role to the host epithelial lining, such as antibiotic use or surgery [1]. This may result in host susceptibility to colonization by pathogens until the normal microbiome is reestablished, which can take several weeks [33]. *S. boulardii* helps to restore the normal microflora in this type of patient and the use of probiotics as modulators of the normal microbiome through colonization during the susceptibility period may work as a surrogate until the normal microbiome is reestablished [34].

### 5.2. Antimicrobial Activity

Antagonism against pathogens can be achieved by colonization and exclusion of pathogens, modulation of metabolic and signaling pathways, production of inhibitory compounds or immune modulation [31]. Competition is one of the main mechanisms associated with probiotic activity against gut pathogens: consumption of nutrients by probiotics results in nutrient limitation for pathogenic organisms [31,35]. On the other hand, the ability of probiotics to grow and colonize the gut can lead to a decrease of the gut pH due to the production of metabolites, leading to stressful conditions for pathogens [35]. A possible role for *S. boulardii* in managing pathogenic activity was associated with a protective effect of *S. boulardii* against pathogenic bacteria in yeast-treated mice, although the mode of action is not associated with a reduction of the pathogenic population [21], as well as with another study which observed a protective effect against *Candida albicans* in a murine model [36].

The production of compounds with antimicrobial activity is yet another major mode of action of probiotics. Several components of the probiotic metabolome, such as organic acids, bacteriocins, hydrogen peroxide, diacetyl, or amines limit the growth of pathogenic bacteria [31]. In particular, bacteriocins play a crucial role in the antimicrobial action of probiotic bacteria, especially *Lactobacillus* spp. [35,37,38,39,40]. As for the production of antimicrobial substances by other probiotics, *S. boulardii* possibly secretes proteins that reduce *Citrobacter rodentium* adhesion to host epithelial cells by modulating virulence factors [41]. It also displays antimicrobial activity by secreting 54-kDa, 63-kDa and 120-kDa proteins that cleave microbial toxins or reduce cAMP levels. *S. boulardii* can block toxin receptors or function as a decoy receptor for toxins. The 54-kDa serine protease produced by *S. boulardii* cleaves toxins A and B from *C. difficile* and the enterocytic receptor to which the toxins bind, which causes inflammation, fluid secretion, mucosal permeability and injury in the intestines [42,43]. Other mechanisms that *S. boulardii* uses against *C. difficile* infection are growth inhibition and decreased toxin production due to secreted factors and stimulation of host mucosal disaccharidase activity [44,45]. Another study refers to the ability of *S. boulardii* to inhibit *Escherichia coli* surface endotoxins by dephosphorylation. A 63-kDa alkaline phosphatase targets the lipopolysaccharide (LPS) and contributes to decreased tumor necrosis factor α (TNF-α) cytokine levels [46]. *S. boulardii* also produces a 120-kDa protein that decreases the chloride secretions stimulated by cholera toxin by reducing cAMP levels [47]. *S. boulardii* is also able to adhere to cholera toxin via its cell wall, thus blocking its toxic effects [48]. Despite these observations, the sequencing of *S. boulardii* genomes did not provide a clear identification of the genes encoding these 54-kDa, 63-kDa and 120-kDa proteins [25]. 

*S. boulardii* also confers protection against the lethal toxin produced by *Bacillus anthracis*. The bacterium causes ulcerative lesions from the jejunum to cecum and uses its toxin to disrupt intestinal epithelium integrity, causing mucosal erosion, ulceration and bleeding [49]. The protective effect of *S. boulardii* is associated with maintenance of barrier function and reduction of harmful physiological responses elicited by the toxin, such as formation of stress fibers [50]. The protective effect is achieved by release of proteases and cleavage of the lethal toxin [50].

Some *S. boulardii* strains are able to produce high concentrations of acetic acid, which was found to exert an inhibitory effect in *E. coli* [22]. In turn, the decrease in pH due to acetic acid production is essential for the antimicrobial activity of short-chain organic acids. The combined effect of high acetic acid concentration and lower pH may be an additional mechanism that makes *S. boulardii* an effective probiotic. Moreover, acetic acid is produced under aerobic conditions by *S. boulardii*. Due to the radial oxygen gradient between the epithelial surface (high oxygen levels) and the center of the gut lumen (low oxygen levels), microorganisms colonizing the epithelial surface have greater availability of oxygen [51]. Since acetic acid is produced under aerobic conditions by *S. boulardii*, its production should be higher near the epithelial surface. During antibiotic treatment and pathogen infection, oxygen concentration also increases in the gastrointestinal tract [52,53], which could support the antimicrobial action of *S. boulardii*. 

### 5.3. Adhesion

In order for the host not to mechanically eliminate the gut microbiome, it is crucial that its components adhere to host surfaces [1,4]. Some probiotics express surface adhesins that mediate the attachment to the mucous layer by recognizing host molecules such as transmembrane proteins (integrins or cadherins) and extracellular matrix components (collagen, fibronectin, laminin or elastin) [1,4]. Probiotics can also influence the production of mucin and the barrier function of the intestine, thus hindering adhesion and consequent invasion of pathogenic microorganisms [54]. 

Mucin is produced by epithelial cells to avert adhesion by pathogenic bacteria, whereas successful probiotics should be able to adhere to the intestinal mucous, as is the case of *S. boulardii* [13,55]. The adhesion of *S. boulardii* to the mucus membrane contributes to reducing the availability of binding sites for pathogens [13]. Five *S. cerevisiae* cell wall proteins (encoded by *CIS3*, *CWP2*, *FKS3*, *PIR3* and *SCW4*) were found to mediate adhesion of the yeast cells to the pathogenic bacteria *E. coli*, *Salmonella enterica* serovar *typhimurium* (*S. typhimurium*) and *Salmonella enterica* serovar *typhi* (*S. typhi*) [55]. Other studies have shown that these bacteria are also bound to *S. boulardii* [55,56,57,58]. Additionally, *S. boulardii* also inhibits *C. difficile* adhesion to epithelial cells and displays inhibitory activity on *Entamoeba histolytica* adhesion to erythrocytes [59,60]. This interaction limits the ability of pathogens to bind directly to the intestinal receptors and proceed with host invasion. In fact, *S. boulardii* hinders epithelium invasion by *S. typhimurium* due to steric hindrance caused by its larger size as compared to bacteria [61]. As *S. boulardii* does not significantly bind to epithelial cells of healthy individuals and is quickly flushed out, pathogens bound to *S. boulardii* are possibly flushed together with the yeast cells [13,31,55]. However, *S. boulardii* does have several flocculation genes required for protection against environmental stress and biofilm formation [11]. The characterization of this gene family in the context of host adhesion and colonization could provide further insight on the probiotic features of *S. boulardii*. 

The ability of *S. boulardii* to bind bacterial pathogens has been associated with the presence of mannose residues in the yeast cell wall [56]. This is a similar mechanism to the previously characterized adhesion of bacterial pathogens to the epithelial surface via mannose residues [62], which is the basis for the addition of exogenous sugars as a strategy to inhibit pathogen adhesion [63]. Cell wall mannan oligosaccharides are a common feature in yeast, but the affinity between *E. coli* and *S. boulardii* is higher than with *S. cerevisiae* [56]. Further investigations revealed that bile salts decrease adhesion of bacteria to yeast cells [55], which can have relevant implications for yeasts as successful probiotics. Accordingly, bile salts also decrease the adhesion of probiotic bacteria to intestine epithelia due to diminished surface hydrophobicity and higher surface potential [64], bolstering how important it is for probiotic microorganisms to evolve adaptation strategies within the host.

### 5.4. Immune Modulation

Metabolites produced by the gut microbiome can perform immunomodulatory and anti-inflammatory functions that stimulate immune cells. This ability arises from the interaction between the probiotics and the epithelial cells, dendritic cell monocytes, macrophages and/or lymphocytes [1,31]. Probiotics also promote enhanced phagocytic activity, cell proliferation and production of secretory immunoglobulins IgA and IgM [65].

*S. boulardii* can modulate immunological function by acting as a stimulant or a pro-inflammatory inhibitor. It is capable of modulating the inflammatory process upon *S. typhimurium* infection by decreasing the levels of the pro-inflammatory molecules such as cytokine interleukin 8 (IL-8), mitogen activated protein (MAP) kinases and the (nuclear factor kappa B) NF-κB signaling pathway [58]. An inhibitory effect of *S. boulardii* over MAP kinases and IL-8 levels upon *C. difficile* infection was also observed [66]. Likewise, *S. boulardii* contributes to increasing the levels of anti-inflammatory cytokines (IL-4 and IL-10) and decreasing pro-inflammatory cytokines (IL-1β) upon infection with *E. coli* and *C. albicans* [67]. On the other hand, *S. boulardii* was associated with increased IgA and IgG levels in serum in response to *C. difficile* toxins A and B [68,69]. *S. boulardii* was also found to attach to the surface of dendritic cells [70] and modulate the expression of toll-like receptors (TLRs) and cytokines [70,71,72]. Moreover, *S. boulardii* also causes the imprisonment of T helper cells in mesenteric lymphatic nodes, reducing inflammation [73].

Another study found that in the early phase of *S. typhimurium* infection, *S. boulardii* induces pro-inflammatory cytokine production (interferon-γ—IFN-γ) and represses the production of anti-inflammatory cytokines (IL-10) in the small intestine, but increases the levels of both cytokines in the cecum [57]. This suggests that *S. boulardii* can differentially modulate immune activity through the GI tract [57]. Overall, probiotics may be able to persistently modulate both the innate and adaptive immune responses either locally or systemically [1,31]. The data from several studies indicates that *S. boulardii* plays a pivotal role in immune modulation against the most common GI tract pathogens.

### 5.5. Trophic Effects

*S. boulardii* is a modulator of enzyme activity required to maintain a healthy gastrointestinal tract. It exerts trophic effects such as stimulation of brush border membrane digestive enzymes and nutrient transporter activity [74]. Several studies have shown a wide array of trophic effects stimulated by *S. boulardii*: brush border sucrase, lactase, and maltase activities [44,75,76,77,78]; iso-maltase activity [78]; glucoamylase and *N*-aminopeptidase activity [76]; leucine-aminopeptidase activity [79]; α,α-trehalase activities in the endoluminal fluid and intestinal mucosa; brush border α-glucosidase [80]; spermine, spermidine and putrescine levels in rat jejunal mucosa [75,77]; adenosine triphosphatase, γ-glutamyl transpeptidase, lipase, and trypsin activities and TNF-α, IL-10, transforming growth factor beta (TGF-β), and secretory IgA [5]; diamine oxidase activities, brush border sodium/glucose cotransporter 1 expression and sodium-dependent glucose uptake [74,77]; GRB2-SHC-CrkII-Ras-GAP-Raf-ERK1,2 transduction pathway in rats and decreased p38 MAPK and NF-κB [81,82,83]. 

Probiotics can modulate short chain fatty acids (SCFA: acetate, propionate, or butyrate) and/or branched-chain fatty acid (BCFA: iso-butyrate, 2-methylbutyrate, or isovalerate) synthesis. SCFAs have a complex role in the physiological and biochemical functions in different tissues (intestine, liver, adipose, muscle and brain). *S. boulardii* assists in reestablishing SCFA levels, which are depressed during disease [84,85]. Acetate and butyrate are major SCFAs in intestinal epithelial cells, playing a role in barrier function, anti-inflammatory and immune modulation pathways [86,87]. A study reported that a short-term treatment (6 days) with *S. boulardii* diminishes the incidence of diarrhea in patients receiving enteral nutrition by increasing SCFA levels, particularly butyrate [84]. SCFAs can also present antimicrobial activity, and a study probing several *S. boulardii* and *S. cerevisiae* strains for inhibitory effects in *E. coli* described the production of acetic acid exclusively by *S. boulardii* as an antimicrobial mechanism [22]. Moreover, acetate also stimulates T regulatory cells, induces mucus secretion gene expression, inhibits proinflammatory cytokine CXCL8 and serves as a substrate for the production of butyrate by the microbiome [22].

The activity of many digestive enzymes (sucrase-iso-maltase, maltase-glucoamylase, lactase-phlorizin hydrolase, alanine aminopeptidase and alkaline phosphatase) and nutrient transporters (sodium-glucose transport proteins) may be induced by polyamines secreted by *S. boulardii* [74]. *S. boulardii* secretes polyamines that promote RNA binding and stabilization and, hence, growth and protein (lactase, maltase, sucrase, among others) synthesis [74]. These molecules are also able to shield lipids from oxidation and boost SCFA activity. Polyamines may also affect kinase activities and external signal transduction pathways, therefore modulating the GRB2-SHC-CrkII-Ras-GAP-Raf-ERK1,2 and the PI3K pathways [74]. They can also aid the generation of specific transcripts by interacting with DNA [74]. All of these polyamine functions lead to a general polyamine-triggered metabolic activation in order to regenerate brush border damage and maturation of enterocytes [74,75,80]. Not only does *S. boulardii* induce the enzymatic activities of lactase-phlorizin hydrolase, α-glucosidases, alkaline phosphatases and aminopeptidases, but it also increases glucose intestinal absorption, one of the products of lactose degradation [74]. Production of lactase by the host, partially stimulated by *S. boulardii*, mediates lactose degradation thus alleviating lactose intolerance.

## 6. *S. boulardii* Safety and Clinical Efficacy

Although many probiotics are documented as safe, common safety issues regarding the use of probiotics include: transfer of antibiotic resistance genes, translocation of live organisms from the intestine to other sites of the body, persistence in the intestine and development of adverse reactions [1]. Most of these concerns have been dismissed when evaluating *S. boulardii* safety. *S. boulardii* is not known to acquire resistance genes, unlike bacterial probiotics such as *Lactobacillus* spp. [88,89]. Animal studies show that there is reduced translocation in the treatment with *S. boulardii* when compared with *S. cerevisiae* [90,91,92]. *S. boulardii* does not persist in the intestine after three to five days after discontinuation of the ingestion, according to pharmacokinetic studies [20]. The data available from 90 clinical trials assessing the efficacy and safety of *S. boulardii* has been thoroughly assessed elsewhere [1]. Randomized and controlled trials clearly show the absence of any serious adverse reactions, while only some presented moderate adverse reactions, such as constipation in patients with *C. difficile* infection [93]. Although fungemia is viewed as a potential problem, there were no fungemia cases reported in clinical trials [1]. *S. boulardii*-associated fungemia was observed in patients with serious co-morbidity factors and central venous catheters, which responded well to fluconazole or amphotericin B therapy [91,94,95,96]. Importantly, *S. cerevisiae*-associated fungemia has a worse prognosis than that caused by *S. boulardii* [97].

Clinical trials have investigated the efficacy of *S. boulardii* in the improvement of several GI conditions’ outcome. This yeast was seen to improve the outcome of several diarrhea diseases, including pediatric diarrhea, antibiotic-associated diarrhea, acute diarrhea, traveler’s diarrhea caused by bacterial, viral or parasites, and enteral nutrition-related diarrhea [1,15,98]. *S. boulardii* also improves the outcome in patients suffering from *H. pylori* or *C. difficile* infections by helping bacteria eradication, preventing relapses, reducing adverse reactions, and reducing treatment-associated diarrhea [1,15,98]. IBD is a prevalent GI tract disorder associated with inflammatory diarrheal diseases such as ulcerative colitis, pouchitis and Crohn’s disease [1]. Clinical trial data points to a possible role of *S. boulardii* in reducing treatment relapses [1,15,98], which are frequent in these conditions, although further studies are required to reach compelling conclusions. Irritable Bowel Syndrome (IBS) symptoms also improve with *S. boulardii* administration. It is a condition frequently characterized by abdominal bloating, abdominal pain, and disturbed intestinal transit. These symptoms were shown to be alleviated in 50% of patients upon *S. boulardii* use [99].

## 7. Conclusions

*S. boulardii* is a probiotic yeast with proven efficacy in the treatment of GI conditions, especially when used as an adjuvant to antibiotic treatment. Present data indicate that the benefits of *S. boulardii* appear to be transient and independent of host gut colonization, differentiating its mode of action from other widely used bacterial probiotics. The absence of colonization appears to correlate with pathogen binding as a mechanism to halt pathogen colonization, rather than competitive exclusion due to yeast adhesion. Genomics studies have contributed to pinpoint distinct genome features that mainly confer on *S. boulardii* the ability to resist host stresses, conferring higher viability through GI passage than observed for other common probiotics. *S. boulardii* also elicits a complex immunomodulatory effect with roles in fine-tuning immunological pathways during pathogen infection or chronic diseases. This yeast also contributes to the homeostasis of the normal microbiome and plays a relevant role in modulating secretory functions by intestinal epithelial cells, thus benefitting nutritional requirements of the host. Overall, *S. boulardii* displays a multifactorial role as a probiotic, with proven efficacy and safety in alleviating the symptomology of a number of GI conditions. However, there is a significant knowledge gap between *S. boulardii* phenotypic effects and the underlying genetic basis, especially when compared to *S. cerevisiae*. What are the 54-kDa, 63-kDA and 120-kDA proteins secreted by *S. boulardii* that cleave microbial toxins or reduce cAMP levels? What are the proteins responsible for higher adhesion of *S. boulardii* to pathogenic bacteria, when compared to *S. cerevisiae*, especially considering the differences in flocculin encoding genes? What are the mechanisms that allow *S. boulardii* to overcome the negative impact of bile salts during host adaptation? What are the proteins or cellular components that mediate immune recognition of *S. boulardii* and modulation of the immune response? These questions remain unanswered. Further research on the genetic basis of *S. boulardii* probiotic activity will certainly increase our understanding of this fascinating yeast, while providing important clues for the selection and optimization of even more powerful probiotic fungi. 

## Figures and Tables

**Figure 1 jof-06-00078-f001:**
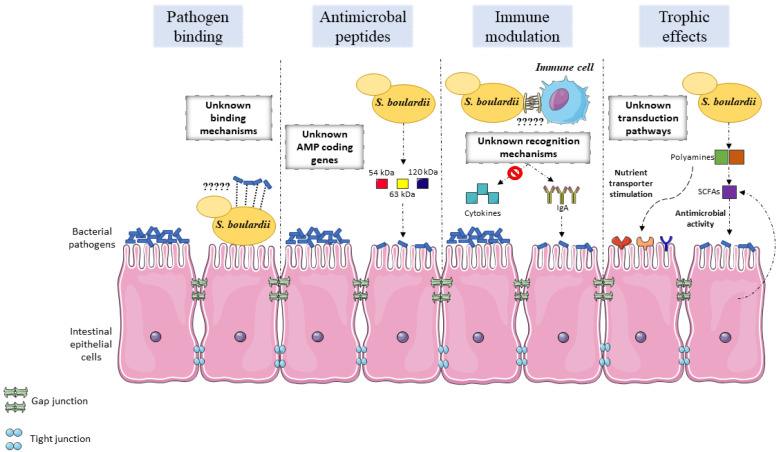
Overview of the main modes of action that support S. boulardii probiotic activity in the intestinal epithelium. Studies have described the outcome of S. boulardii administration in pathogen exclusion, antimicrobial properties, immune modulation, and trophic effects. The genetic basis and mechanistic details that underlie these observations are not fully understood and their clarification could be key to better exploit this yeast and how to potentiate general probiotic activity. Pathogen exclusion is mainly achieved by pathogen binding to the yeast cells, rather than competition for epithelial binding sites with the pathogens. The yeast cell wall components responsible for the binding, the correspondent pathogenic receptors and the binding dynamics have not been fully investigated. Antimicrobial action is achieved, at least partially, by the secretion of still unknown proteins with antimicrobial effects. The genes that code for these proteins have not been identified and could provide further clues on the mode of action of S. boulardii. Immune modulation and the effect of S. boulardii on inflammatory pathways has been uncovered to some extent. The mechanistic insights and dynamics of S. boulardii interaction with immune cells still need to be ascertained to better understand the yeast action in the immunological function. Multiple trophic effects have been described to be stimulated by S. boulardii on intestinal epithelial cells. Some pathways have been elucidated, although the multitude of trophic effects suggests concerted action and crosstalk between yeast and host cell sensory pathways.

**Table 1 jof-06-00078-t001:** Metabolic, physiological and genetic features of *S. cerevisiae* and *S. boulardii*. The data shown was collected from several studies [11,12,13,16,17,18,19,20,21].

Features		*S. Cerevisiae*	*S. Boulardii*
Optimal growth temperature [12]		30 °C	37 °C
High temperature resistance (52 °C) [12]		45% viability	65% viability
Acid pH resistance (pH = 2 for one hour) [12,18]		No—30% viability	Yes—75% viability
Tolerance to bile acids (>0.3%(*w*/*v*)) [12]		No—Survival up to 0.15% (*w*/*v*)	No—Survival up to 0.10% (*w*/*v*)
Basic pH resistance (pH = 8) [12,18]		Yes	Yes
Assimilation of galactose [16,17,19]		Yes	No
Ploidy [18]		Diploid or haploid	Always diploid
Homo or heterothallic [11]		Homothallic	Homothallic
Mating type [13]		Both	Both
Sporulation [16,18]		Sporogenous	Asporogenous, but produces fertile hybrids with *S. cerevisiae*
Pseudo-hyphal switching [18]		Normal	Increased
Retrotransposon (Ty elements) [11]		Intact Ty elements	No intact Ty1, 3 or 4 elements
Adhesion to epithelial cells	Normal microbiome (mice and human) [18,20]	No	No
	Gnotobiotic mice [21]	Unknown	Yes
	Humans treated with ampicillin [20]	Unknown	Yes

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
