# Peer review of "Saccharomyces boulardii: What Makes It Tick as Successful Probiotic?"

_jof, 2020, doi:10.3390/jof6020078_

Round 1

Reviewer 1 Report

Dear authors, I highly recommend the manuscript "Saccharomyces boulardii: what makes it tick as successful probiotic?" by authors  Miguel C. Teixeira, Pedro Pais, Vanda Almeida and Melike Yilmaz to be published in its present form. I have no negative comments at all.

Author Response

Thank you for your encouraging comments.

Reviewer 2 Report

I am satisfied with this your interesting review and suppose that it can be published in its present form without any revision

Author Response

(The authors gave the same response as above.)

Reviewer 3 Report

The review article by Pais and colleagues provides an overview of the use of S. boulardii as a probiotic. They also provide comparative characteristics of S. boulardii and S. cerevisiae.

This review is very well organized. The introduction and abstract are well-written and clearly structured. I think the manuscript is acceptable after revising minor points.

Minor comments:

Table 1. Italicate "S. cerevisiae" and "S. boulardii" in table caption. Also, add reference to each data (ex. 45% viability [11])

Lines 89-93  Please add reference(s).

Line 93  I guess "Saccharomyces sensu strico" should be italic.

Line 94  Use smaller font for "L-". 

Lines 142-143  Please name some probiotics that are resistant / non-resistant to hydrolyzed acids and bile salts.

Lines 146-155  It may be helpful to the reader to cite the number of days or hours S. boulardii survived the GI tract, conditions, intestinal environments, or low pH, etc.

Line 165  "Caco-2 cell"

Figure 1  Please enlarge the figure to increase the resolution.

Lines 235, 247, and 411  "kDa" (e.x. 100-kDa protein)

Author Response

Thank you for your constructive remarks. We have done our best to addressed them all. Specifically:

Table 1. Italicate "S. cerevisiae" and "S. boulardii" in table caption. Also, add reference to each data (ex. 45% viability [11])

R: Species names are now in italic and the correspondent references for each feature in Table 1 have been added.

Lines 89-93  Please add reference(s).

R: The reference supporting these sentences is now featured in the text.

Line 93  I guess "Saccharomyces sensu strico" should be italic.

R: Saccharomyces sensu stricto is now in italic.

Line 94  Use smaller font for "L-".

R: A smaller font is now being used.

Lines 142-143  Please name some probiotics that are resistant / non-resistant to hydrolyzed acids and bile salts.

R: Examples of probiotics that are resistant or non-resistant to stressors present in the gastrointestinal system are now featured in the text. This information is now present in lines 143-148.

Lines 146-155  It may be helpful to the reader to cite the number of days or hours S. boulardii survived the GI tract, conditions, intestinal environments, or low pH, etc.

R: Temporal information on the viability of S. boulardii under the conditions mentioned in the text is now present in the manuscript. This information is now present in lines 149-161.

Line 165  "Caco-2 cell".

R: The text was corrected to read “Caco-2 cells”. Now in line 171.

Figure 1  Please enlarge the figure to increase the resolution.

R: The figure has been enlarged.

Lines 235, 247, and 411  "kDa" (e.x. 100-kDa protein)

R: The text now reads 54/63/120-kDa proteins. The correction was implemented in multiple instances through lines 240-254.